# Birds as Environmental Bioindicators of Genotoxicity in Brazilian Cerrado Farmlands: An In Situ Approach

**DOI:** 10.3390/ani15213208

**Published:** 2025-11-04

**Authors:** Henrique Nazareth Souto, Edimar Olegário de Campos Júnior, Marcos Vinicius Bohrer Monteiro Siqueira, Carlos Fernando Campos, Cassio Resende Morais, Boscolli Barbosa Pereira, Sandra Morelli

**Affiliations:** 1Institute of Genetics and Biochemistry, Universidade Federal de Uberlândia, Uberlândia 38405-319, MG, Brazil; henriquenazareth@gmail.com (H.N.S.); carllosfernando@hotmail.com (C.F.C.); cassio.1015@hotmail.com (C.R.M.); morelli@ufu.br (S.M.); 2Institute of Biological Sciences, Universidade Federal de Juiz de Fora, Juiz de Fora 36036-900, MG, Brazil; 3Campus of Frutal, Universidade do Estado de Minas Gerais-Unidade Frutal, Frutal 38202-436, MG, Brazil; marcos.siqueira@uemg.br; 4Institute of Geography, Universidade Federal de Uberlândia, Uberlândia 38405-319, MG, Brazil; boscolli@ufu.br

**Keywords:** cytogenetic biomarkers, micronucleus, organochlorine pesticides, sentinel birds

## Abstract

**Simple Summary:**

Birds are excellent environmental sentinels because they can reflect pollution effects that may go unnoticed in other organisms. In this study, we used wild birds living on coffee farms in the Brazilian Cerrado to detect genetic damage caused by pesticides. Blood samples from 152 birds were analyzed using a cytogenetic test that identifies small fragments of DNA damage called micronuclei. Birds living on farms that used more pesticide mixtures showed higher levels of micronuclei than those from a preserved reference site. Some species, such as the Gray-fronted Dove (*Leptotila rufaxilla*), Blue-black Grassquit (*Volatinia jacarina*), and Rufous-bellied Thrush (*Turdus rufiventris*), were especially sensitive, indicating their potential use as early-warning bioindicators. These findings demonstrate that monitoring wild birds can help identify harmful levels of pesticide exposure in agricultural landscapes, contributing to safer farming practices and environmental conservation in tropical regions.

**Abstract:**

Birds have played a crucial role as environmental monitors throughout history, ranging from the use of canaries to detect methane and carbon monoxide in mines to the decline of raptors and seabirds during the DDT era due to widespread organochlorine pesticide contamination. Owing to their high diversity and capacity for bioaccumulation, birds are widely recognized as effective indicators of environmental change and pollutant exposure. Cytogenetic techniques have been increasingly applied over the past two decades to assess micronuclei formation resulting from interactions with clastogenic and aneugenic chemical compounds. The main goals of this study were (a) to evaluate a subset of the bird community in the southeastern Brazilian Cerrado as potential environmental indicators of pesticide exposure using the erythrocyte micronucleus test and (b) to investigate possible associations between bird morphometric traits and micronuclei frequency. Birds were sampled from three groups of coffee farms in the Brazilian Cerrado. Blood samples were collected from 152 individuals (122 on farms and 30 at the reference site) via the metatarsal vein, followed by slide preparation for micronucleus analysis. Two slides were prepared per bird; each slide was scored for 10,000 erythrocytes, and MN frequency was reported as the mean across slides. The species *Leptotila rufaxilla*, *Volatinia jacarina*, *Galbula ruficauda*, *Gnorimopsar chopi*, *Molothrus bonariensis*, *Passer domesticus*, *Turdus leucomelas*, and *Turdus rufiventris* exhibited six or more micronuclei per 10,000 erythrocytes, indicating the highest potential as bioindicators of environmental contamination. Micronuclei frequency in erythrocytes was positively correlated with the use of mixed pesticides, with variation depending on the size of the coffee farms. Although a slight negative biological trend was observed between micronuclei frequency and certain morphometric traits, particularly bill length, no statistically significant correlations were found. Similarly, birds from large farms exhibited a slight reduction in certain morphometric features, though these differences were also not statistically significant. These results highlight the utility of selected bird species as early-warning bioindicators for pesticide exposure in tropical agroecosystems.

## 1. Introduction

Bioindicators are widely employed to assess environmental conditions in various fields, including ecology, toxicology, pollution control, agriculture, forestry, and wildlife management [1]. The primary advantages of using organisms as indicator species include their relative ease of identification, straightforward measurement procedures, and low associated costs [2].

Birds have long been recognized as effective environmental bioindicators due to their high sensitivity to ecological disturbances and their ability to reflect contamination levels across multiple biological scales, from cellular alterations to population-level effects [3,4,5,6,7]. This has been demonstrated in multiple contexts, such as the clastogenic effects of food storage in poultry [8], detection of urban pollutants [9], monitoring of Cerrado fragments [10,11], pesticide-related pollution [12], and evaluation of ecotoxicological endpoints [13].

The integration of cytogenetic tools into environmental monitoring has significantly enhanced the utility of birds as bioindicators. Techniques such as the erythrocyte micronucleus test, evaluation of nuclear abnormalities (e.g., binucleated cells, nuclear buds, and nucleoplasmic bridges), and other biomarkers of cellular stress enable the detection of chromosomal damage and genetic instability caused by environmental pollutants [14]. These cytogenetic biomarkers provide valuable insights into the health and genetic integrity of sentinel species [15,16].

The micronucleus test (MNT), a cytogenetic assay developed by [17,18], has been widely applied to evaluate clastogenic and aneugenic effects, initially in mammals. However, recent studies have successfully employed this assay to assess the genotoxic effects of environmental contaminants in birds, particularly heavy metals. The MNT in birds provides a validated cytogenetic endpoint sensitive to clastogenic and aneugenic agents. In nucleated avian erythrocytes, micronuclei represent chromosomal fragments or whole chromosomes excluded from the main nucleus during erythropoiesis. This assay yields comparable baselines across species and enables robust in situ screening of genotoxic exposure in agroecosystems [10,19,20,21,22].

Combining cytogenetic tools with traditional bioindicator approaches, such as population and community surveys, physiological and behavioral assessments, biotic indices, chemical tissue analyses, and bioassays, improves the accuracy and reliability of environmental monitoring, especially in detecting pesticide-related effects in avian fauna [23].

Pesticides are synthetic chemical compounds broadly used to control disease vectors, agricultural pests, weeds, and other organisms considered harmful to human activity [24]. Their widespread use in rural and urban areas results in chronic exposure and potential health risks. In humans, pesticide exposure typically occurs through ingestion of contaminated food, inhalation, or dermal contact [25].

In birds, exposure to pesticides may be accidental, when birds are not the intended targets, or intentional, such as when poisoned baits are used to eliminate predators [26]. Carbamate pesticides, for instance, have been associated with mortality in birds [26]. Additionally, pesticides can induce a variety of sublethal effects, including cellular damage, metabolic disruption, reproductive and morphological abnormalities, and genotoxicity [27]. These effects may include alterations to eggshell structure [28], DNA damage, and increased micronucleus frequency in erythrocytes [12,29].

The Brazilian Cerrado is recognized as a global biodiversity hotspot, but faces growing threats from agricultural expansion and pesticide-intensive monocultures, particularly in coffee plantations [30]. Despite the advantages of avian bioindicators, field-based monitoring in tropical regions such as the Cerrado remains limited due to logistical challenges and a lack of standardized cytogenetic baselines for wild bird species [31]. By integrating morphometric assessments with cytogenetic biomarkers, this study offers a comprehensive and integrative approach to understanding the sublethal impacts of pesticide exposure on avian communities within tropical agroecosystems.

Because seasonality, landscape mosaic and short-distance movements can influence capture probabilities and exposure histories, we combined independent farms with a nearby reference site in the same ecoregion to mitigate confounding and to anchor cytogenetic baselines. Accordingly, the objectives of this study were: (1) to assess selected bird communities in the Southeastern Brazilian Cerrado as environmental indicators of pesticide exposure through erythrocyte micronucleus testing; and (2) to explore possible associations between bird morphometric traits and micronucleus frequency.

## 2. Materials and Methods

### 2.1. Study Areas

Monte Carmelo, in the state of Minas Gerais, Brazil, was selected as the study site because of its regional economic importance in coffee production and its representativeness of pesticide-intensive agricultural practices in the Brazilian Cerrado. The municipality combines a long-standing tradition of conventional coffee farming with patches of relatively well-preserved native vegetation [32]. This landscape mosaic enables the simultaneous investigation of anthropogenic impacts and baseline ecological conditions within the same ecoregion, making it a strategic location for ecotoxicological studies using avian bioindicators.

The study was conducted on nine coffee farms (Figure 1), located in the surroundings of Monte Carmelo (18°47′56.98″ S; 47°19′3.64″ W). Farms were classified by productive area as small (≤20 ha), medium (20–50 ha), and large (>50 ha). The average coffee yield across all farms was approximately 50 sacks per hectare (sacks/ha), regardless of total productive area. The farms were distributed across distances ranging from 3 to 30 km from one another, ensuring that each location served as an independent sampling unit with distinct exposure conditions.

All sampled farms maintained a mosaic of native Cerrado and pasture areas, adopting conventional cultivation techniques for *Coffea arabica*, including chemical fertilization, regular pesticide application, and either mechanized or manual harvesting [33]. All farms hold Rainforest Alliance certification, indicating compliance with standardized environmental practices and pesticide protocols. Despite this certification, a variety of pesticides are applied to coffee crops, following a gradient pattern: larger farms apply greater volumes and a broader diversity of chemical compounds.

The same classes of agrochemicals are used across all farms, with minor variation in active ingredients, concentrations, and application frequencies. Pesticides are applied either manually or mechanically; large farms typically rely on mechanized spraying, which may increase airborne pesticide dispersion. The total volume of pesticide syrup applied—standardized in liters per hectare (L/ha)—was 959.5 L/ha on small farms, 1209.5 L/ha on medium farms, and 2210 L/ha on large farms. A detailed breakdown of active ingredients, concentrations, and total volumes per farm category is provided in Table 1.

Additionally, four sampling points (U1 to U4) located at the edge of urban areas were selected to monitor bird movements between agricultural and urban environments. However, none of the 122 birds marked on farms were observed in these urban-edge zones, suggesting limited displacement during the study period.

A reference site (negative control area) comprising approximately 130 ha of preserved native Cerrado vegetation within the same ecoregion was used for baseline cytogenetic comparisons. This site is consistently referred to as the reference site throughout the manuscript. A prior survey by [10], conducted in four Cerrado fragments, recorded 93 bird species—approximately four times the species richness observed, on average, on individual study farms —underscoring the high ecological integrity of the control area. Although the reference site shares the same biome, differences in ecological composition and levels of anthropogenic disturbance justify its use as a comparative control.

The climate in Monte Carmelo is classified as tropical savanna (Aw), according to the Köppen–Geiger system, with a rainy season from October to April and a dry season from May to September [34]. Field sampling was conducted in both seasons to account for seasonal variability in bird activity and pesticide use, which typically follows agricultural cycles in the region.

### 2.2. Data Collection (Birds)

Authorization was obtained from the Brazilian Institute of Environment and Renewable Natural Resources (IBAMA, protocol no. 50528-1) and approval was granted by the Ethics Committee on the Use of Animals in Research (CEUA/UFU, protocol no. 120/15), wild birds were sampled using mist nets measuring 12 m × 3 m. The nets were deployed from 6:00 a.m. to 5:00 p.m., from January to the end of October 2021. Sampling was systematic, with up to seven individuals collected per species among those most frequently captured. All birds were identified to the species level following the taxonomic classification proposed by [35].

Marked individuals received a small dot of non-toxic paint (using a colour scheme to indicate farm type: green for small farms, yellow for medium farms, and red for large farms) on the dorsal upper back to enable short-term resighting without affecting behavior or thermoregulation. Forty hours of visual observations were conducted across all sampling points to detect marked individuals. No bird marked at a given site was observed at any other site during the study period, indicating strong site fidelity. This marking method followed the technique proposed by [36] to assess site fidelity and local residence. Species nomenclature followed the standards of the Brazilian Committee of Ornithological Records [37].

### 2.3. Blood Samples, Micronuclei Tests, and Staining

A total of 122 farm birds had blood samples collected by puncturing the metatarsal vein with a sterile needle. At the reference site, an additional 30 individuals were sampled. One to two drops of blood were immediately smeared on microscope slides. Two duplicate slides were prepared per bird. Slides were prepared directly in the field and later processed in the laboratory following [38]. These individuals belonged to the same species recorded on farms, including *Columbina talpacoti*, *Crotophaga ani*, *Guira guira*, *Dacnis cayana*, *Volatinia jacarina*, *Elaenia flavogaster*, *Myiozetetes similis*, *Tyrannus melancholicus*, *Galbula ruficauda*, *Gnorimopsar chopi*, *Molothrus bonariensis*, *Mimus saturninus*, *Passer domesticus*, *Thamnophilus doliatus*, *Thamnophilus torquatus*, *Turdus leucomelas*, *Turdus rufiventris* and *Zonotrichia capensis*. One slide per bird was prepared directly in the field.

All slides were transported to the Cytogenetics Laboratory of the Federal University of Uberlândia (UFU), Minas Gerais, Brazil, for processing. Slides were fixed with absolute methanol for 10 min, followed by staining with 5% aqueous Giemsa solution for 10 min, and rinsed with distilled water, following the protocol described by [38].

### 2.4. Evaluation of Micronuclei

Micronucleus (MN) analysis was conducted on peripheral blood smears prepared from nucleated erythrocytes using a binocular light microscope (Olympus Corporation, Tokyo, Japan) equipped with a 100× oil immersion objective. All micronucleus values are expressed as MN per 10,000 erythrocytes, computed as the mean of duplicate slides per individual.

The identification of MN followed the criteria established by [39], whereby a structure was classified as a micronucleus only if: (a) it was clearly separated from the main nucleus without any chromatin connection; (b) it was located in the same focal plane and exhibited similar staining intensity and texture to the primary nucleus; and (c) its diameter ranged between one-tenth and one-third of the main nucleus.

To ensure consistency and minimize bias, all readings were performed blindly by a single trained observer, without knowledge of sample origin. No re-readings were performed, but the use of duplicate slides per bird provided internal consistency to the evaluation. The frequency of micronucleated erythrocytes was expressed as the number of MN per 10,000 cells, allowing direct comparisons across sites and groups.

### 2.5. Statistical Analysis

Statistical analyses were performed using GraphPad Prism 7.0 (GraphPad Software Inc., San Diego, CA, USA). Data normality was assessed with the Shapiro–Wilk test. As the MN frequency data did not conform to a normal distribution, non-parametric statistical tests were applied.

Differences in MN frequencies among bird groups from small, medium, and large farms, as well as the reference site, were evaluated using the Kruskal–Wallis test, followed by Dunn’s post hoc test for pairwise comparisons. To examine the association between MN frequency and pesticide application intensity (L/ha), Spearman’s rank correlation coefficient was calculated.

Statistical significance was set at *p* < 0.05. Results are presented as medians with interquartile ranges (IQR), which is appropriate for non-parametric datasets. We quantified the association between MN per 10,000 erythrocytes and pesticide application intensity using Spearman’s rank correlation with exact *p*-values. To assess robustness, 10,000 random-label permutations provided a null distribution for r. We also report 95% CIs for r using bias-corrected and accelerated bootstrap resampling at the farm level.

## 3. Results

Photographs of four representative bird species commonly observed in the study areas—*Galbula ruficauda*, *Leptotila rufaxilla*, *Volatinia jacarina*, and *Turdus rufiventris*—are shown in Figure 2.

A total of 152 individuals from 21 bird species were sampled across coffee farms and the reference site (122 on farms, 30 at the reference site). Micronucleus frequency in erythrocytes ranged between 0 and 8 per 10,000 cells. At the reference site, the mean MN frequency was 0.26 ± 0.52 per 10,000 erythrocytes. In contrast, birds captured on farms exhibited variable frequencies, with MN detected in a subset of individuals, predominantly in medium and large farms MN was detected in 32 of 122 farm individuals (26.2%). High MN values (≥5 per 10,000 erythrocytes) occurred in 14 of 122 individuals (11.5%), predominantly in medium and large farms.

Median MN frequencies were 0 for the control area, 1 for small farms, 3 for medium farms, and 5 for large farms. Kruskal–Wallis analysis confirmed significant differences among groups (H = 70.87; *p* < 0.0001), with post hoc tests revealing that birds from medium and large farms differed significantly from the control group (*p* < 0.05).

MN was positively associated with pesticide application intensity across farms (Spearman r = 0.71, 95% CI [0.18, 0.93], *p* = 0.03; permutation *p* = 0.032), indicating a moderate-to-strong monotonic relationship (Figure 3). Taken together, pesticide application records and cytogenetic outcomes indicate a coherent gradient: application volumes rise from small to large farms (Table 1), and MN medians increase in the same order (reference site = 0; small = 1; medium = 3; large = 5). This pattern is consistent across species captured in more than one farm-size category.

Several species showed notably elevated MN frequencies, including *Leptotila rufaxilla* (granivore-frugivore, ground forager), *Volatinia jacarina* (granivore-insectivore), *Galbula ruficauda* (insectivore), *Gnorimopsar chopi* (omnivore), *Molothrus bonariensis* (granivore-insectivore), *Passer domesticus* (omnivore), *Turdus leucomelas* (frugivore-insectivore), and *Turdus rufiventris* (frugivore-insectivore), which presented six or more MN per 10,000 cells in one or more individuals (Table 2). Although individual variation was present, these species tended to be more affected than others.

To facilitate interspecific comparison, we highlight candidate bioindicator species with elevated MN per 10,000 erythrocytes and broad occurrence on medium and large farms: *Galbula ruficauda*, *Leptotila rufaxilla*, *Volatinia jacarina*, *Molothrus bonariensis*, *Passer domesticus*, *Turdus leucomelas*, and *Turdus rufiventris*. This pattern is consistent with dietary exposure pathways, as insectivores and omnivores are likely to ingest contaminated prey, and granivores (e.g., *Volatinia jacarina*, *Leptotila rufaxilla*) may consume seed residues or soil particles. Guild assignments and bill metrics are shown in Table 3.

No significant correlations were detected between MN and morphometric parameters, indicating that interspecific differences and dietary exposure are more likely to drive MN variability than body size. A slight negative trend with bill length was observed, but it was not statistically significant. Spearman analyses across all morphometric measures showed weak or non-significant associations with MN frequency (Figure 4).

Similarly, birds from large farms exhibited a slight reduction in some morphometric features, such as bill length and weight, though once again, these differences were not statistically significant. Comparative analysis of these traits across farm size categories is illustrated in Figure 5.

Application records indicated higher spray volumes and a broader mix of active ingredients in large farms relative to small and medium farms. This gradient, coupled with mechanized spraying that increases airborne drift, provides a parsimonious explanation for the higher MN values observed in birds captured in large farms.

## 4. Discussion

The results of this study indicate that various bird species can potentially serve as effective bioindicators of environmental genotoxicity. While some individuals presented zero micronuclei (MN), this may reflect an efficient physiological system for removing altered erythrocytes rather than a true absence of genotoxic effects [40]. Species with higher MN frequencies, such as *Galbula ruficauda*, *Leptotila rufaxilla*, *Volatinia jacarina*, and *Turdus rufiventris*, demonstrated particular promise as biomonitoring candidates due to their consistent presence across sites and elevated cytogenetic responses [41,42].

Surprisingly, we found no significant correlation between MN frequency and bird morphometric traits such as body mass, wing length, or tail length. Although not statistically significant, variations in bill length may suggest high MN levels. This suggests a possible link between dietary habits and genotoxic exposure, particularly in birds that forage on small grains or insects, which are more likely to accumulate pesticide residues [12]. These findings align with ecotoxicological studies across vertebrate classes, including fish [43], amphibians [44], reptiles [42], and even humans [45,46], reinforcing the sensitivity and applicability of the micronucleus test in environmental assessments.

The vulnerability of feeding guilds was further confirmed in this study. Insectivorous and granivorous species appear more susceptible to genotoxic agents, likely due to trophic-level pesticide exposure, either by consuming contaminated insects or grains or by foraging in soils treated with agrochemicals. This trophic exposure pathway highlights the importance of integrating feeding ecology into biomonitoring frameworks and helps explain interspecific variability in MN frequency observed even within the same habitat type [12].

Diet can mediate exposure through two non-exclusive pathways. Insectivores may accumulate residues by consuming contaminated arthropods and via trophic transfer, including contact with sprayed foliage and drift during peak application periods [27,29]. Granivores may ingest seed-coating residues and soil particles during ground foraging, particularly in simplified matrices and along mechanized spray tracks [13,23]. These routes are consistent with the higher MN values observed in insectivorous and granivorous species on medium and large farms, in line with previous field-based avian biomonitoring in agricultural settings [12]. Species such as *Galbula ruficauda and Volatinia jacarina* illustrate these pathways, the former as an insectivorous forager frequently contacting treated foliage and the latter as a granivore with potential ingestion of seed residues and soil particles.

Multiple independent lines of evidence converge on an exposure–response gradient: (i) farm-level application volumes are highest in large farms; (ii) MN medians increase from small to large farms; and (iii) site fidelity of marked individuals minimizes movement-driven dilution of exposure. This convergence supports an interpretation centered on local pesticide loads rather than on body size or sampling artifacts.

Although birds from large farms showed significantly higher MN frequencies than those from smaller farms and the control site, these cytogenetic effects did not result in statistically significant morphometric alterations. One hypothesis is that energy demands associated with the removal of damaged cells may compete with developmental processes, leading to subtle anatomical changes over time [47]. However, these effects may require chronic exposure over longer periods to become detectable. It is also worth noting that the baseline frequency of MN in avian erythrocytes is typically lower than in mammals or fish, which requires species-specific thresholds for accurate genotoxic assessments [48,49]. The combined effect of higher application volumes and mechanized dispersion plausibly increases contact rates with residues across foraging microhabitats, reinforcing the observed MN gradient with farm size.

Sex determination was not included as a variable in this study because field procedures required the rapid handling and release of birds immediately after capture, minimizing stress and ensuring compliance with ethical protocols. This constraint, combined with the fact that most Cerrado bird species lack evident sexual dimorphism, rendered reliable sexing unfeasible without invasive or molecular methods. Species such as *Leptotila rufaxilla*, *Elaenia flavogaster*, *Guira guira*, and *Crotophaga ani* exhibit virtually identical plumage between males and females. Furthermore, previous cytogenetic studies in wild birds [15,16,41] have reported no significant influence of sex or body mass on micronucleus frequency, indicating that MN variability is primarily driven by ecological exposure rather than intrinsic biological traits. Therefore, potential sex-related bias in the present dataset is considered minimal.

This study contributes to a growing body of literature emphasizing the utility of wild birds in environmental biomonitoring. Earlier works focused on single-species models, such as gulls or parrots [8,48,49,50], whereas recent studies have expanded this approach to encompass avian communities [10,11,12]. Our findings support the argument that bird diversity enhances biomonitoring robustness by capturing interspecific differences in exposure and sensitivity to environmental stressors.

Additionally, the results of this study reinforce the need for caution in interpreting cytogenetic responses as direct reflections of health risks across species. Baseline MN values may vary due to physiological or ecological traits, and sample sizes are often insufficient to draw species-specific conclusions. Moreover, field studies like ours are inherently limited by the unpredictable nature of species capture, which may lead to uneven representation across habitats. As such, proposals for bioindicator species must be evaluated for feasibility and consistency across monitoring efforts. The need for large sampling efforts and the possibility of species absence at a given site can compromise indicator reliability as bioindicators in long-term monitoring frameworks.

For applied monitoring we recommend sampling aligned with peak spraying periods; duplicate slides per individual, reporting MN per 10,000; selection of common resident species with high capture probability; and systematic reporting of a nearby reference site. We also encourage archiving raw slide-level counts to support meta-analyses.

Finally, this work aligns with global efforts to assess and mitigate the impacts of intensive agricultural practices on biodiversity. The strong correlation between MN frequency and pesticide use highlights the cytogenetic burden imposed by large-scale monocultures and reinforces the relevance of Sustainable Development Goals (SDGs), especially SDG 15 (Life on Land) and SDG 12 (Responsible Consumption and Production). As recently emphasized by [13,23], integrating biomonitoring indicators into national biodiversity strategies is essential for mitigating the silent erosion of ecosystem services caused by pollution and habitat simplification.

## 5. Conclusions

The frequency of micronuclei in erythrocytes (MN per 10,000) was positively correlated with a mixture of pesticides used on coffee farms, in a manner dependent on farm size. Although there was a slight negative biological trend between MN frequency and certain morphometric traits, particularly bill length, these associations were not statistically significant. Similarly, birds from larger farms exhibited a subtle reduction in some morphometric measurements, but again, no statistically significant differences were detected. These findings support the integration of cytogenetic biomarkers into environmental monitoring protocols within coffee-producing landscapes.

## Figures and Tables

**Figure 1 animals-15-03208-f001:**
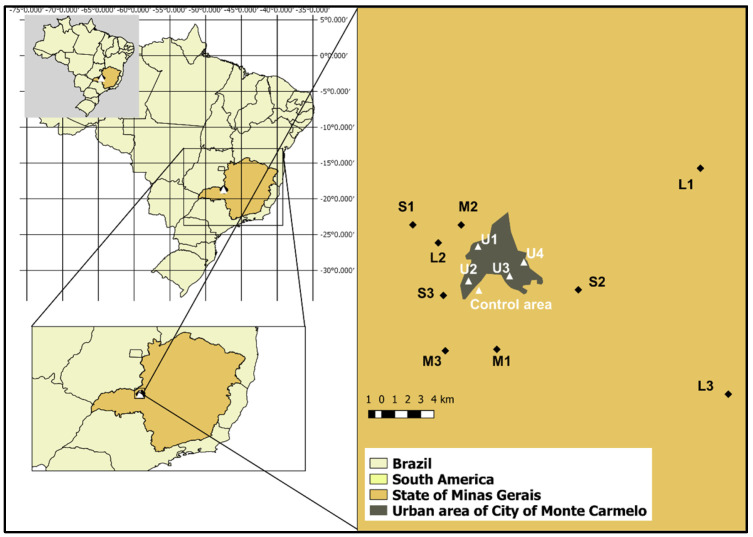
Geographic location of study sites near Monte Carmelo, Minas Gerais, Brazil. Nine coffee farms were selected based on productive capacity and classified into three categories: small (S1–S3), medium (M1–M3), and large (L1–L3). Four urban observation points (U1–U4) were established at the edges of the urban zone to monitor bird movement across rural–urban boundaries. The Negative Control Area represents a preserved Cerrado fragment used as a reference site for comparative cytogenetic analysis. Insets show the location within Brazil and the State of Minas Gerais.

**Figure 2 animals-15-03208-f002:**
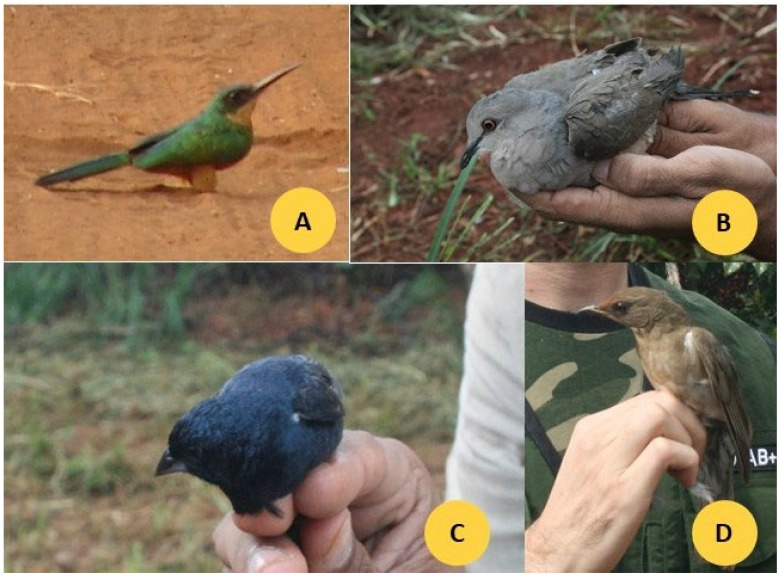
Photographs of four species ((**A**) *G. ruficauda*, (**B**) *L. rufaxilla*, (**C**) *V. jacarina*, (**D**) *T. rufiventris*) found in the farms (Monte Carmelo, Minas Gerais, Brazil), very frequently.

**Figure 3 animals-15-03208-f003:**
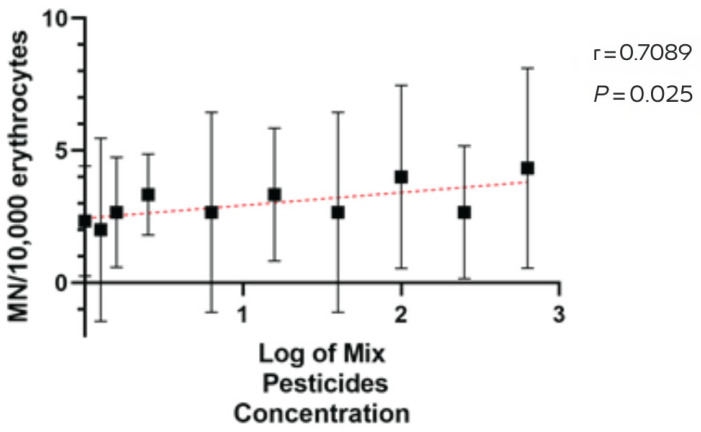
Spearman correlation between pesticide application intensity (log scale) and micronuclei (MN) per 10,000 erythrocytes across farms. Black squares show farm-level central tendency with 95% confidence intervals (vertical bars). The red dashed line depicts the fitted monotonic trend for visualization. Spearman r = 0.71 (95% CI 0.18–0.93).

**Figure 4 animals-15-03208-f004:**
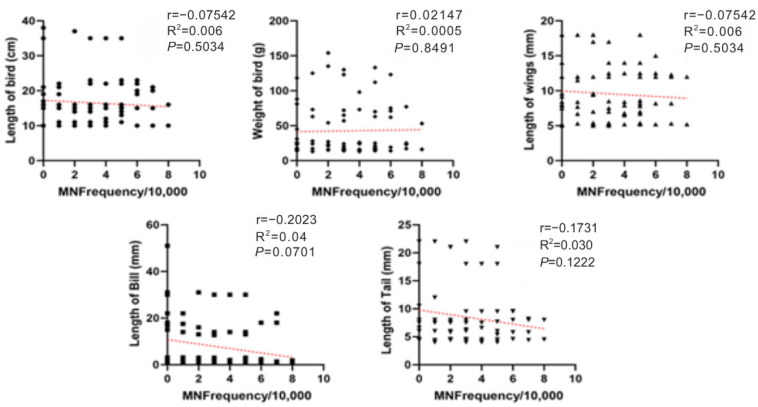
Spearman correlations between MN frequency (MN per 10,000 erythrocytes) and avian morphometrics (body length, body mass, wing length, bill length, and tail length) for birds sampled at farms in Monte Carmelo, Minas Gerais, Brazil. Points are individual birds. Different point shapes distinguish categories within each panel solely to improve readability and have no analytical meaning. The red dashed line denotes the fitted trend in each panel. Reported statistics are Spearman r and *p*-values; R^2^ refers to the linear fit where shown.

**Figure 5 animals-15-03208-f005:**
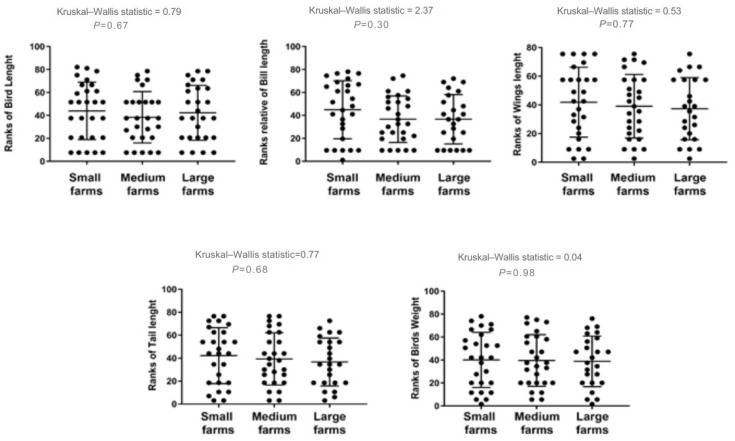
Differences in anatomical measurements for birds sampled on farms of different sizes in Monte Carmelo, Minas Gerais, Brazil. Variables were rank-transformed prior to the Kruskal–Wallis test (y-axes show ranks). Dots represent individual birds; the horizontal line is the group median and the whiskers show the interquartile range (IQR). The Kruskal–Wallis statistic and the corresponding P-value are shown above each panel.

**Table 1 animals-15-03208-t001:** Active ingredient and its units of concentration applied in the farms at Monte Carmelo, Minas Gerais, Brazil.

Active Ingredient	Concentration (g/L)	Class of Pesticide	Pesticide Syrup Volume (L/ha)		
			Small Farms	Medium Farms	Large Farms
Copper hydroxide	691	Fungicide	0.5	0.5	1
Carbendazim	250	Fungicide	5	5	5
Pyraclostrobin	133	Fungicide	500	500	500
Imidacloprid	700	Insecticide	50	100	100
Spiromesifen	240	Insecticide/Acaricide	400	600	1600
Chlorantraniliprole	350	Insecticide	4	4	4
		Total	959.5	1209.5	2210

**Table 2 animals-15-03208-t002:** Micronucleus frequency in birds captured on farms of different sizes in Monte Carmelo, Minas Gerais, Brazil. Values are MN per 10,000 erythrocytes (mean ± SD). Farm totals sum to 122 individuals. The reference site sample comprised 30 additional individuals.

Family	Scientific Name	Common Name	*n*	MN per 10,000 Erythrocytes (Mean ± SD)		
				Small Farms	Medium Farms	Large Farms
Columbidae	*Columbina talpacoti*	Ruddy Ground Dove	1	0	-	-
	*Leptotila rufaxilla*	Grey-fronted Dove	9	-	-	6 ± 0
Cuculidae	*Crotophaga ani*	Smooth-billed Ani	3	0	3 ± 0	4.5 ± 0.7
	*Guira guira*	Guira Cuckoo	3	1.5 ± 0.7	1.67 ± 1.52	5 ± 0
Thraupidae	*Dacnis cayana*	Blue Dacnis	2	1 ± 1.41	5 ± 0	-
	*Tangara sayaca*	Sayaca Tanager	3	2 ± 0	0	-
	*Tersina viridis*	Swallow Tanager	1	-	2 ± 0	-
	*Volatinia jacarina*	Blue-black Grassquit	18 **	1.86 ± 1.07	2.43 ± 1.27	5.57 ± 1.62
Tyrannidae	*Elaenia flavogaster*	Yellow-bellied Elaenia	1	0	0	4 ± 0
	*Myiozetetes similis*	Social Flycatcher	4	2 ± 1	4 ± 0	4.5 ± 0.7
	*Tyrannus melancholicus*	Suiriri Flycatcher	3	1 ± 0	3.5 ± 0.7	4.5 ± 0.7
Galbulidae	*Galbula ruficauda*	Rufous-tailed Jacamar	17 **	1 ± 0	0	7 ± 0
Icteridae	*Gnorimopsar chopi*	Chopi Blackbird	9	0.1	-	6.5 ± 0.7
	*Molothrus bonariensis*	Shiny Cowbird	15 **	2 ± 0	3 ± 0	8 ± 0
Mimidae	*Mimus saturninus*	Chalk Mockingbird	1	-	1 ± 0	-
Passeridae	*Passer domesticus*	House Sparrow	5	-	6 ± 0	6 ± 1.41
Thamnophilidae	*Thamnophilus doliatus*	Barred Antshrike	3	0	-	3 ± 0
	*Thamnophilus torquatus*	Rufous Antshrike	1	1 ± 0	-	-
Turdidae	*Turdus leucomelas*	Pale-breasted Thrush	4	3 ± 0	5 ± 0	6 ± 0
	*Turdus rufiventris*	Rufous-bellied Thrush	16 **	4 ± 0	3 ± 0	4.5 ± 0.7
Passerellidae	*Zonotrichia capensis*	Rufous-collared Sparrow	3	0	2.5 ± 0.7	5 ± 0
TOTAL			122	1.38 ± 1.20	2.59 ± 1.65 *	5.31 ± 5.40 *
Reference site (multiple species)			30	0.26 ± 0.52		

* Statistical difference in the groups when compared to the reference site; ** Viable species for potential biomonitors (with at least 5 individuals per point).

**Table 3 animals-15-03208-t003:** Trophic guild of bird species and bird bill lengths collected in the farms at Monte Carmelo, Minas Gerais, Brazil. This table summarizes ecological traits and bill metrics.

Family	Scientific Name	Common Name	Guild	Bill Length (mm)		
				*Small farms*	*Medium farms*	*Large farms*
Columbidae	*Columbina talpacoti*	Ruddy Ground Dove	Granivorous	15.0	-	
	*Leptotila rufaxilla*	Grey-fronted Dove	Granivorous	-	-	24.0
Cuculidae	*Crotophaga ani*	Smooth-billed Ani	Omnivorous	35.0	34.0	35.0
	*Guira guira*	Guira Cuckoo	Omnivorous	36.5	36.0	36.0
Thraupidae	*Dacnis cayana*	Blue Dacnis	Omnivorous	31.0	13.0	-
	*Tangara sayaca*	Sayaca Tanager	Frugivorous	21.0		-
	*Tersina viridis*	Swallow Tanager	Omnivorous		15.0	
	*Volatinia jacarina*	Blue-black Grassquit	Granivorous	10.3	10.3	10.1
Tyrannidae	*Elaenia flavogaster*	Yellow-bellied Elaenia	Omnivorous	18.0	17.0	16.0
	*Myiozetetes similis*	Social Flycatcher	Insectivorous	17.3	16.5	
	*Tyrannus melancholicus*	Suiriri Flycatcher	Omnivorous	21.5	21.0	19.5
Galbulidae	*Galbula ruficauda*	Rufous-tailed Jacamar	Insectivorous	22.0	20.0	19.0
Icteridae	*Gnorimopsar chopi*	Chopi Blackbird	Omnivorous	17.8		23.5
	*Molothrus bonariensis*	Shiny Cowbird	Omnivorous	18.0	16.0	16.0
Mimidae	*Mimus saturninus*	Chalk-browed Mockingbird	Omnivorous	-	26.0	-
Passeridae	*Passer domesticus*	House Sparrow	Omnivorous	-	14.0	14.5
Thamnophilidae	*Thamnophilus doliatus*	Barred Antshrike	Insectivorous	20.0	-	17.0
	*Thamnophilus torquatus*	Rufous-winged Antshrike	Insectivorous	-	21.0	-
Turdidae	*Turdus leucomelas*	Pale-breasted Thrush	Omnivorous	23.0	22.0	22.0
	*Turdus rufiventris*	Rufous-bellied Thrush	Omnivorous	25.0	21.0	21.0
Passerellidae	*Zonotrichia capensis*	Rufous-collared Sparrow	Granivorous	17.0	14.0	13.0

## Data Availability

The original contributions presented in this study are included in the article. Underlying raw data involve confidential farm records and are not publicly available; further inquiries can be directed to the corresponding author.

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
