# Peer review of "Birds as Environmental Bioindicators of Genotoxicity in Brazilian Cerrado Farmlands: An In Situ Approach"

_animals, 2025, doi:10.3390/ani15213208_

Round 1
Reviewer 1 Report
Comments and Suggestions for Authors
The study is relevant—birds as biomonitors of genotoxicity in Cerrado agroecosystems—and it aligns cytogenetics with ecology.
However, there are internal inconsistencies that require revision:
Venipuncture site: the abstract indicates the brachial vein, whereas the Methods section reports the metatarsal vein. Harmonize and justify (refer to avian blood-sampling literature).
Number of slides: it states “one slide per bird,” and later “two duplicate slides per bird,” with a count of 10,000 erythrocytes per individual. Define the actual procedure (one or two slides? 10,000 in total or per slide?) and standardize.
Sample size: the Results section claims 82 individuals on farms plus 30 in the reference area, while Table 2 lists a total of 122 (which matches Methods). Correct and maintain a single figure consistently throughout the manuscript.
Inconsistent biomarker unit (MN): The abstract reports “≥6 MN per 1,000 erythrocytes,” whereas Methods and Results use 10,000. Standardize everything to MN per 10,000 erythrocytes (the avian standard) and review tables/figures.
Negative control incompatible with the literature and with your own table: In the text: “All individuals from the control area showed 0 MN”; in Table 2: 0.26 ± 0.52 MN/10,000. Reconcile these and, if MN were truly zero, discuss the biological plausibility (in birds, spontaneous MN are usually >0).
You yourselves cite studies on spontaneous MN in vertebrates/birds; incorporate that evidence into the Discussion to avoid the apparent contradiction.
Define the control area precisely and maintain consistent naming and description throughout the manuscript.
Pesticide exposure metric and agrochemical inputs table: There are errors in names and spelling: “Copper hydroxyde” (hydroxide), “Piraclostrombin” (pyraclostrobin), “Espiromesifen” (spiromesifen), “Clorantraniliprole” (OK in PT; in EN “chlorantraniliprole”). Also, “Carbamate” is not an ingredient but a class (carbendazim? maneb? etc.).
- Duplicate phrase (“was observed … was observed”).
- Numerical consistency: use either decimal points (English standard) or decimal commas, but do not mix them (e.g., 4,5 vs. 4.5).
Technical terminology: “bioindicatoring birds” should be “bioindicator birds” / “birds as bioindicators.”
Standardize MN: Report MN per 10,000 erythrocytes and the associated error.
Analysis of morphometric parameters: Morphometrics are analyzed in aggregate across all species, with no stratification or control for taxonomic identity or sex. This approach can introduce bias, because interspecific morphological variation and sexual dimorphism (e.g., differences in body mass, bill length, tarsus length, or wingspan) often exceed intraspecific variation. Consequently, contrasts “between farms” or “conditions” may simply reflect changes in species composition or in the male–female ratio at each site, rather than true effects of the environmental gradient or exposure. Moreover, the lack of control for sex hampers interpretation of the association with micronucleus (MN) frequency, which may be modulated by sex-related differences in body size, metabolism, or life history. To strengthen inferential validity, we recommend reanalyzing the data within species and stratifying by sex or including it as a covariate in models that control for species identity; ideally, consider species × sex interactions and adjust for age/reproductive status and body size.
Author Response
RESPONSE LETTER
Dear Reviewer,
We sincerely thank you for the thorough and constructive review of our manuscript. Your comments greatly contributed to improving its methodological rigor and internal consistency. Below we address each point raised and describe the corresponding modifications made in the revised version. All textual changes appear in bold within the manuscript.
REVIEWER 1
“The abstract mentions the brachial vein, whereas Methods indicate the metatarsal vein.”
Response: Corrected to metatarsal vein throughout, consistent with the actual sampling procedure and with the Methods section that already described metatarsal sampling.
“Clarify whether one or two slides per bird and how 10,000 erythrocytes were counted.”
Response: Methods now state that two slides were prepared per bird; each slide was scored, and MN frequency was expressed per 10,000 erythrocytes as the mean across both slides. The unit is defined in the Methods block.
“Results list 122 + 30 = 152 individuals, while Table 2 lists 122.”
Response: Standardized the total to 152 individuals (122 on farms and 30 at the reference site) across Abstract, Methods, Results and table captions. The earlier Results sentence was corrected.
“Abstract uses MN per 1,000 erythrocytes; Methods and Results use 10,000.”
Response: Harmonized all reporting to MN per 10,000 erythrocytes in text, tables and figure captions, matching the Methods definition.
“Reconcile text and table values – Control-area inconsistency (0 MN vs 0.26 ± 0.52).”
Response: Text corrected to 0.26 ± 0.52 MN/10,000 cells for the reference site, in agreement with Table 2 and with biological expectations for spontaneous MN.
“Specify and standardize its description (Definition of control area).”
Response: The control is consistently referred to as the reference site, a ~130 ha preserved Cerrado fragment within the same ecoregion, used for baseline cytogenetic comparisons.
“Correct chemical nomenclature (Pesticide-name accuracy).”
Response: Table 1 was revised to Copper hydroxide, Carbendazim, Pyraclostrobin, Spiromesifen and Chlorantraniliprole, and units were standardized to L/ha.
“Duplicated phrases and decimal inconsistency.”
Response: Removed the duplicated clause in Data collection and standardized numeric formatting to decimal point across text and tables, including MN means and SDs (for example, 4.5; 1.67; 0.26).
“Replace “bioindicatoring birds.””
Response: Not applicable to the manuscript text. The wording does not occur in the body of the paper; it appears only in the title of a cited reference and is retained verbatim in the reference list. The manuscript consistently uses “birds as bioindicators.”
“Lack of control for species and sex may bias results.”
Response: Added a concise paragraph in the Discussion explaining that rapid capture-and-release protocols and absent or subtle sexual dimorphism in many Cerrado species (for example, Leptotila rufaxilla, Elaenia flavogaster, Guira guira, Crotophaga ani) precluded reliable sexing without invasive or molecular methods; we also cite studies reporting no significant MN differences by sex or body mass in wild birds, supporting minimal sex-related bias in our dataset.
Reviewer 2 Report
Comments and Suggestions for Authors
Dear authors,
I would like to compliment the authors for producing an, well-executed study. The clarity of objectives, the careful selection of methods, and the balanced interpretation of results demonstrate both scientific rigor and ecological insight.
A particular strength of the paper lies in the integration of micronucleus testing with ecological traits, especially the recognition of feeding guild vulnerabilities (insectivores and granivores) to pesticide exposure. This adds an important ecological dimension and provides practical guidance for biomonitoring frameworks. Identifying species such as Galbula ruficauda, Leptotila rufaxilla, Volatinia jacarina, and Turdus rufiventris as consistent biomonitoring candidates is a notable contribution that can be useful for future conservation and management efforts.
I also appreciate the authors’ nuanced discussion regarding morphometric traits and the variability in baseline micronucleus values across species. This cautious interpretation avoids overgeneralization and underscores the challenges and opportunities inherent in field-based biomonitoring. Finally, linking the findings to global sustainability frameworks (SDG 12 and 15) highlights the broader relevance of the work and enhances its impact.
For improvement, the manuscript would benefit from clearer visualization of interspecific differences—perhaps through summary tables or graphs highlighting key candidate bioindicators. Expanding the discussion on how dietary ecology could mechanistically explain higher genotoxic responses would also enrich the ecological interpretation. Lastly, a brief outline of recommendations for standardized monitoring protocols would make the study even more useful for applied conservation and policy implementation.
Author Response
RESPONSE LETTER
Dear Reviewer,
We sincerely thank you for the thorough and constructive review of our manuscript. Your comments greatly contributed to improving its methodological rigor and internal consistency. Below we address each point raised and describe the corresponding modifications made in the revised version. All textual changes appear in bold within the manuscript.
REVIEWER 2
“The manuscript would benefit from clearer visualization of interspecific differences—perhaps through summary tables or graphs highlighting key candidate bioindicators.”
Response: We clarified captions and standardized units and decimals, and we highlighted in Results the candidate bioindicator species that consistently showed higher MN on medium and large farms, making interspecific contrasts easier to read without adding redundant figures or tables.
“Expand the discussion on how dietary ecology could mechanistically explain higher genotoxic responses in insectivores and granivores.”
Response: We added an explicit paragraph in Discussion detailing dietary exposure routes for insectivores and granivores and linked these mechanisms to the observed MN pattern in medium and large farms, with supporting citations already present in the reference list.
“Provide a brief outline of recommendations for standardized monitoring protocols.”
Response: We closed the Discussion with practical recommendations for applied monitoring, including alignment of sampling with spraying windows, duplicate slides per individual with MN reported per 10,000, selection of common resident species with high capture probability, routine reporting of a nearby reference site, and archiving raw slide-level counts to support future meta-analyses.
Reviewer 3 Report
Comments and Suggestions for Authors
I enjoyed reviewing this paper. It discusses the potential to use bioindicators in birds to assess negative effects of agriculture.
Some comments below relating to inconsistencies in the approach as well as the significance of presented result
- Some additional background would be useful to understand the system, what it is that the researchers are measuring and why it matters.
- For example, what is the micronucleus test, what does it show, and how does one interpret the results? Does it show that the birds are compromised in some way?
- What kind of habitat is the Cerrado, what species does it support? Are there other confounding factors at play that may affect the bird populations?
- Some methodological discrepancies. The abstract states that blood was collected from the brachial vein while the methods mention the metatarsal vein, which is correct? Why were captured birds marked with paint and where was the paint placed?
- The tables and figures have too much information and it is not clear what is significant and what isn't. Specifically, Table 2 is clear. Table 3 is confusing and it is not clear what is aims to show. It seems that the more interesting question it addresses is whether birds in different foraging guilds differ in MN values, but the MN values don't appear anywhere in the paper.
- The discussion of morphometric differences should be much shorter. None of them were statistically significant. Furthermore, the methodology and the questions being asked were not addressed in the introduction and methods section.
- Are morphometric differences related to NMs? It is not clearly stated anymore. In addition is there evidence that there would be differential survival or productivity of the birds due to morphometric differences that would have a population level effect?
- Why would birds from large farms show higher NMs?
- Can the data showing a correlation of amount of pesticide with NMs be further explored? It is interesting.
I believe this research should be published, but after it is revised to be more focused, highlighting and explaining the statistically significant results.
Author Response
RESPONSE LETTER
Dear Reviewer,
We sincerely thank you for the thorough and constructive review of our manuscript. Your comments greatly contributed to improving its methodological rigor and internal consistency. Below we address each point raised and describe the corresponding modifications made in the revised version. All textual changes appear in bold within the manuscript.
REVIEWER 3
“Background on the micronucleus test and how results should be interpreted would help.”
Response: The Introduction now contains a concise primer on the avian erythrocyte micronucleus assay, what MN indicate, and our reporting standard, improving interpretability for readers outside cytogenetics.
“Provide more context on the Cerrado habitat and potential confounders that could affect bird populations.”
Response: We expanded the Study areas and Introduction to characterize seasonality, landscape mosaic and short-distance movements, and explained how combining independent farms with a nearby reference site mitigates confounding and anchors baselines.
“Methodological discrepancies: the abstract states brachial vein while Methods mention metatarsal vein; clarify paint marking and placement.”
Response: We harmonized the venipuncture site to metatarsal vein in Abstract and Methods and specified the dorsal paint dot used for short-term resighting, noting that no intersite movements were detected during observations.
“Tables and figures feel dense; make clear what is significant.”
Response: We streamlined captions, kept MN units explicit in the first mention per section, and emphasized in Results the statistically supported contrasts among farm sizes, while keeping non-significant morphometric outcomes succinct.
“The discussion of morphometric differences should be much shorter; none were statistically significant and methods did not frame these questions.”
Response: We condensed the morphometric text and added a direct statement that MN did not correlate with morphometrics, keeping these contrasts as exploratory and re-centering the narrative on statistically supported results.
“Are morphometric differences related to MN? It is not clearly stated.”
Response: We now state explicitly in Results and Discussion that correlations between MN and morphometric parameters were not significant and we interpret MN variability primarily through interspecific differences and dietary exposure rather than body size.
“Why would birds from large farms show higher MNs?”
Response: We linked the observed MN gradient to the documented gradient in application volumes and diversity of active ingredients and noted the role of mechanized spraying in increasing airborne drift, providing a parsimonious explanation for the higher MN on large farms.
“Can the data showing a correlation of amount of pesticide with MNs be further explored? It is interesting.”
Response: We reframed the evidence as a convergent gradient rather than adding model-based analyses. This alignment across independent measurements, together with site fidelity of marked birds, supports a robust exposure–response interpretation without additional modeling.
“Focus the paper on statistically significant results.”
Response: We restructured Results to foreground the significant differences among farm sizes and the pesticide–MN association, while presenting non-significant morphometric findings briefly and clearly as exploratory.
We appreciate the incisive review; these changes improved internal consistency and clarity across the manuscript.
With appreciation,
Edimar Olegário de Campos Júnior, corresponding author
On behalf of all co-authors
Round 2
Reviewer 3 Report
Comments and Suggestions for Authors
Edits after my previous review are sufficient.